# Diabetes induced decreases in PKA signaling in cardiomyocytes: The role of insulin

**Craig A. Eyster[1], Satoshi Matsuzaki[1], Maria F. Newhardt[1,2], Jennifer R. Giorgione[1], Kenneth M. Humphries[1,2] ***

1 Aging and Metabolism Research Program, Oklahoma Medical Research Foundation, Oklahoma City, Oklahoma, United States of America, 2 Department of Biochemistry and Molecular Biology, University of Oklahoma Health Sciences Center, Oklahoma City, Oklahoma, United States of America

* Kenneth-Humphries@omrf.org

**Data Availability Statement:** All relevant data are within the manuscript and its Supporting Information files.

**Funding:** This work was supported by National Institutes of Health Grant R01HL125625 (to K. M.

## Abstract

The cAMP-dependent protein kinase (PKA) signaling pathway is the primary means by which the heart regulates moment-to-moment changes in contractility and metabolism. We have previously found that PKA signaling is dysfunctional in the diabetic heart, yet the underlying mechanisms are not fully understood. The objective of this study was to determine if decreased insulin signaling contributes to a dysfunctional PKA response. To do so, we isolated adult cardiomyocytes (ACMs) from wild type and Akita type 1 diabetic mice. ACMs were cultured in the presence or absence of insulin and PKA signaling was visualized by immunofluorescence microscopy using an antibody that recognizes proteins specifically phosphorylated by PKA. We found significant decreases in proteins phosphorylated by PKA in wild type ACMs cultured in the absence of insulin. PKA substrate phosphorylation was decreased in Akita ACMs, as compared to wild type, and unresponsive to the effects of insulin. The decrease in PKA signaling was observed regardless of whether the kinase was stimulated with a beta-agonist, a cell-permeable cAMP analog, or with phosphodiesterase inhibitors. PKA content was unaffected, suggesting that the decrease in PKA signaling may be occurring by the loss of specific PKA substrates. Phospho-specific antibodies were used to discern which potential substrates may be sensitive to the loss of insulin. Contractile proteins were phosphorylated similarly in wild type and Akita ACMs regardless of insulin. However, phosphorylation of the glycolytic regulator, PFK-2, was significantly decreased in an insulin-dependent manner in wild type ACMs and in an insulin-independent manner in Akita ACMs. These results demonstrate a defect in PKA activation in the diabetic heart, mediated in part by deficient insulin signaling, that results in an abnormal activation of a primary metabolic regulator.

## Introduction

Heart disease is the leading cause of death for patients with type I or type II diabetes [1]. This is in part because diabetes directly impacts cardiac function independently of other comorbidities. This is termed diabetic cardiomyopathy and it is a multi-factorial condition resulting

H.). The funders had no role in study design, data collection and analysis, decision to publish, or preparation of the manuscript.

**Competing interests:** The authors have declared that no competing interests exist.

from the metabolic stresses of disrupted insulin signaling, hyperglycemia and hyperlipidemia, and mitochondrial dysfunction [2]. In addition, there are also disruptions in protein kinase A (PKA) signaling, the molecular pathway that mediates the metabolic and contractile responses to sympathetic stimulation [3, 4]. While the molecular mechanisms contributing to diabetic cardiomyopathy are highly interrelated, the relationship between metabolic perturbances and changes in PKA signaling are not fully understood.

In the healthy heart the sympathetic nervous system functions through β-adrenergic signaling to increase cardiac contractility. Catecholamines bind to $G\alpha_s$-coupled β-adrenergic receptors, stimulate adenylate cyclase, and subsequently increase cAMP to activate PKA. PKA then phosphorylates proteins involved in calcium cycling (troponin, SERCA, and phospholamban) and proteins that affect metabolic substrate selection (phosphofructokinase-2 (PFK-2) and acetyl-CoA carboxylase-2) [5, 6]. Glucose uptake and oxidation are the primary means of meeting the rapid increase in energy demands in response to sympathetic stimulation [6, 7]. In this way the increase in contractility is orchestrated with activation of metabolic pathways to ensure energy demands are met.

As an insulin sensitive tissue, the heart is affected by either decreases in circulating insulin or by the loss of insulin signaling that occur with type 1 or type 2 diabetes [8, 9]. The primary role of insulin is to increase glucose uptake and metabolism. Thus, the decrease in insulin signaling contributes to the metabolic inflexibility whereby the heart increases reliance on fatty acid oxidation, at the expense of decreased glucose usage, to meet energetic demands [10]. Over the long term, this metabolic inflexibility promotes lipotoxicity, mitochondrial dysfunction, and oxidative stress. Increasing evidence suggests there are interactions between insulin and β-adrenergic signaling. For example, hyperinsulinemia can blunt PKA signaling via an increase in phosphodiesterase 4 which increases cAMP hydrolysis [11, 12]. In our own work, we found PKA signaling is affected in a type 1 diabetic mouse model via changes in PKA activity that are downstream of receptor activation and adenylate cyclase activity [3]. Furthermore, we identified that a loss of insulin signaling, in both type 1 and type 2 diabetic conditions, decreases the content of the PKA substrate, PFK-2 [4]. In the healthy heart, phosphorylation of PFK-2 increases the production of fructose-2,6-bisphosphate, an allosteric activator of PFK-1 which is a committed and rate-limiting step of glycolysis [6]. Thus, the loss of insulin signaling disrupts a mechanism whereby β-adrenergic signaling increases glycolysis to meet energetic demands.

The goal of the present work was to define how the loss of insulin signaling impacts β-adrenergic signaling in cardiomyocytes. Adult mouse cardiomyocytes (ACMs) were isolated from control and Akita diabetic mice and then cultured in the presence or absence of insulin. ACMs were subsequently stimulated with β-adrenergic agonists and PKA signaling was determined by immunofluorescence microscopy. We have identified a striking decrease in PKA signaling in wild type ACMs cultured in the absence of insulin. This effect was mirrored in ACMs isolated from Akita type 1 diabetic mice, regardless of the presence of added insulin. Using phospho-specific antibodies, we found that the phosphorylation of proteins involved in calcium regulation were unaffected by the absence of insulin. In contrast, the metabolic target, PFK-2, was highly sensitive. Our results demonstrate the effects of PKA on cardiomyocyte function is dependent upon the actions of insulin.

## Materials and methods

### Adult mouse cardiomyocyte isolation

Adult cardiomyocytes from 5-month C57BL/6J or C57BL/6J-$Ins2^{Akita}$/J male mice (Akita, The Jackson Laboratory 003548) were isolated and cultured as previously described [3, 13]. Akita

mice are a well-established model of hypoinsulinemia and hyperglycemia [14]. Blood glucose was measured by a glucose test strip (Contour) at the time of sacrifice to confirm hyperglycemia. All Akita mice had blood glucose levels of at least 400 mg/dl. Briefly, after isoflurane administration the heart was excised, the aorta was cannulated, and it was then perfused with type II collagenase (Worthington #LS004176). Calcium was reintroduced to the subsequent single cell suspension and cells were plated on laminin (Corning 354232) coated plates. Media was switched to serum-free culture media (minimal essential medium with Hanks' balanced salt solution, Gibco (11575–032) supplemented with 0.2mg/mL sodium bicarbonate, penicillin-G, 0.1%BSA, glutamine, 10mM butanedione monoxime, and 5μg/mL insulin as indicated. Cells were cultured 18h at 37˚C and 5%$CO_2$ with indicated drugs as described in figure legends. All procedures were approved by the Oklahoma Medical Research Foundation Animal Care and Use Committee.

## Antibodies and drugs

Rabbit polyclonal antibodies to phospho-PKA substrate (9621S), PKA C-α (4782S), phospho-PFK2 (13064S), phospho-Ser16/Thr17-phospholamban (8496S), and phospho-Troponin I were purchased from Cell Signaling Technology. Rabbit polyclonal anti-PDE4D (ab14613) was purchased from Abcam. Alexa Fluor 488 goat anti-rabbit IgG (A11034) and Alexa Fluor 546 phalloidin (A22283) were purchase from Invitrogen. Insulin solution human (19278), (-)-Isoproterenol hydrochloride (16504), 3-Isobutyl-1-methylxanthine (15879) and 8-Bromoadenosine 3',5'-cyclic monophosphate sodium salt (B7880) were purchased from Sigma. Phosphodiesterase inhibitor Tocriset containing Milrinone, Cilostamide, Zardaverine, (R)-(-)-Rolipram, and Ro 20–1724 (Cat. No. 1881) along with MMPX (Cat. No. 0552) and EHNA hydrochloride (Cat. No. 1261) were purchased from Tocris.

## Microscopy

Methods for immunofluorescent staining have been previously described [15] and adapted for primary mouse cardiomyocytes (ACMs). Briefly, ACMs were plated on laminin coated coverslips (Fisherbrand Microscope Cover Glass, 12-545-80) (1 coverslip per well, 24-well plate) for 1h post isolation. Cells were cultured overnight and treated with drugs as described in the figure legends. Following incubation, cells were washed 1X with PBS (Gibco 14190–144) and fixed for 20min in 4% paraformaldehyde (Electron Microscopy Sciences 15710). Cells were washed 2X with PBS and blocked for 1 hr in 2% Blocker BSA (Thermo 37525). Coverslips were inverted onto 50μL of block solution containing 0.1% Triton X-100 (Sigma T9284) and 1–250 dilution of primary antibodies as indicated on parafilm covered 150mm gridded tissue culture dish (Falcon 353025) and incubated overnight at 4˚C. Coverslips were returned to tissue culture dish and washed 3X with block solution and then inverted on 50μL of block solution containing .1% Triton X-100 (Sigma T9284) and 1–250 dilution of secondary antibodies/phalloidin for 1hr at room temperature. Coverslips were then washed 2X with block solution and 1X with PBS and then inverted onto 4μL Vectashield mounting media with DAPI (Vector Laboratories H-1200) and sealed with nail polish (Electron Microscopy Sciences 72180). Cells were imaged on a Zeiss LSM-710 confocal (Carl Zeiss). The microscopy settings were kept the same between each experiment to facilitate unbiased comparisons of fluorescence intensities. Micrographs are maximum intensity projections of 12 picture z-stacks average step size of 1.5μm. Projection images were quantitated in Zen Black (Carl Zeiss version 2012 SP5 FP3) from maximum intensity projection by drawing a free polygon outline around the cell and measuring mean fluorescence intensity. Each experiment constitutes of at least three biological

replicates, indicated in Figure Legends, with at least six individual cells per experiment for a total of at least eighteen total cells quantitated for each data point.

## Western blot analysis

Cardiomyocytes were cultured in 12 well plates, with or without insulin, and drugs were added as indicated in the figure legends. Media was then removed, cells were washed with 0.5 mL PBS, and 75 uL of 1X sample buffer containing 25 mM DTT and 1X Halt Protease/Phosphatase Inhibitor Cocktail (ThermoFisher #78442) was added per well. Samples were heated at 95°C for 5 min, resolved by SDS-PAGE (4–12% NuPAGE Bis-Tris gel, Thermo Fisher), transferred to nitrocellulose membranes, and blocked for 30 min with Odyssey TBS blocking buffer (LI-COR). Antibodies were diluted 1:2000 in block buffer and added to blots overnight at 4°C, subsequently washed the following day, and the secondary antibody (IRDye 800CW, LI-COR; 1:10,000 dilution) was incubated for 1h. Following additional washing, blots were imaged on an Odyssey CLx system and analyzed using the Image Studio software (LI-COR).

## PKA activity

PKA activity was assayed using the PepTag Non-Radioactive cAMP-Dependent Protein Kinase Assay Kit (Promega) as previously described [16]. Briefly, ACMs were isolated and cultured for 18hr in the presence or absence of insulin. ACMs were then treated with 0.25μM isoproterenol for 30min. Media was removed and ACMs were lysed on ice with 100μL RIPA buffer (100mM KCL, 100mM $NaPO_4$, 0.1% NP40, PH 7.4) containing HALT phosphatase and protease inhibitor cocktail (ThermoFisher) and 10mM IBMX (Sigma). Following protocol guidelines, reaction buffer, PepTag A1 peptide, and water were premixed. Next, 20μL of freshly prepared lysate (at approximately 0.1mg/ml of total protein) was added and incubated for 15 min at RT. Conditions were optimized to ensure the reaction rate was linear at this time point. The reaction was stopped by the addition of 1mM cAMP Dependent Protein Kinase Inhibitor peptide (PKI-tide; Sigma) in a 50% glycerol solution to a final concentration of 45uM. Phosphorylated peptide was separated from unphosphorylated peptide by electrophoresis on a .8% agarose gel (100 V for 30 min). The gel was imaged with a D-Digit scanner (LI-COR) and the intensities of phosphorylated peptides were analyzed using ImageJ software (National Institutes of Health, Bethesda, MD). The protein concentration of each lysate was determined by Bradford assay and was used to standardize activities.

## Statistical analysis

GraphPad Prism 7.02 was used for statistical analysis and mean fluorescence intensity values were evaluated using one-way ANOVA with multiple comparisons using Tukey's test. Statistical analysis was performed on the total number of cells analyzed, comprised of at least 18 cells from 3 unique cardiomyocyte preparations and is indicated in the Figure Legends. Similar statistical significance and the same conclusions are reached if instead the data from a given cell preparation are averaged and then analyzed. Statistical significance is noted in the figure legends.

## Results

### β-adrenergic signaling is decreased under diabetic conditions in cardiomyocytes

Initial experiments were performed to validate methodology for evaluating PKA signaling by immunofluorescence in adult mouse cardiomyocytes (ACMs). ACMs were isolated from control mice and cultured for 18h in the presence of insulin and then stimulated with

isoproterenol (ISO, 0.25μM) for 30min. Cells were then fixed and PKA activity was visualized by immunofluorescence microscopy using an antibody that specifically recognizes the protein consensus phosphorylation sequence (RRXS/T, where S or T is phosphorylated) that is specific for PKA substrates [5]. This antibody is widely used in the literature (123 citations per CiteAb. com) and has been previously used as a means to identify changes in PKA activity by immuno-histochemistry [17]. We observed low levels of PKA substrate phosphorylation basally and this was increased approximately 3-fold by ISO treatment (Fig 1A, top panels). Detection of PKA activity by this manner revealed largely diffuse staining but with increased intensity proximal to the sarcolemma and intermittently in areas consistent with Z-bands.

We next examined how the lack of insulin affects PKA signaling. Freshly isolated ACMs were cultured in insulin-free media for 18h and then stimulated with ISO. PKA-substrate phosphorylation was significantly blunted both basally and following ISO treatment (Fig 1A and 1B). This decrease in PKA-substrate phosphorylation was not due to altered kinetics. A time course study, with increasing durations of ISO stimulation, revealed phosphorylation reached a maximal threshold within 10min regardless of whether insulin was present (S1 Fig). In contrast to the immunofluorescence data, no significant differences were observed when PKA substrate phosphorylation was examined by Western blot (Fig 1C and 1D). It is possible that less abundantly expressed proteins may be differentially phosphorylated by the presence or absence of insulin but not detected by Western blot. We also measured PKA activity directly (Fig 1E) and found that culturing ACMs without insulin decreased basal PKA activity. However, the activation of PKA by ISO was not significantly affected by the insulin status. This supports that the loss of immunofluorescence intensity with the PKA substrate antibody is not due to overt loss of PKA activity. Rather, unique epitopes may be detected by the antibody under conditions that maintain native protein confirmations, as with immunofluorescence detection, as compared to the denaturing conditions of SDS-PAGE.

We next examined whether the chronic hypoinsulinemia that occurs with type 1 diabetes is also associated with changes in PKA signaling. Akita mice develop type I diabetes in the absence of obesity and insulitis and this is mediated by a mutation in the Ins2 gene that results in its improper release in response to glucose [14, 18]. Akita ACMs were isolated and cultured overnight in the presence or absence of insulin. As shown in Fig 1A and 1B, Akita ACMs had substantially reduced PKA substrate phosphorylation upon stimulation with ISO as compared to WT. Furthermore, insulin did not enhance ISO-stimulated PKA substrate phosphorylation in Akita ACMs. This supports that decreased insulin signaling affects PKA signaling and that overnight insulin treatment of Akita ACMs is insufficient for rescue.

## Insulin signaling is necessary downstream of cAMP production

The decrease in PKA substrate phosphorylation in cardiomyocytes cultured without insulin may be due to changes in β-adrenergic receptors or their response to ligand binding, thereby leading to a dampened response to ISO stimulation. We therefore examined direct activation of PKA using the cell permeable cAMP analog, 8Br-cAMP. Like ISO, 8Br-cAMP induced a robust increase in PKA-substrate phosphorylation in ACMs isolated from control mice and cultured with insulin. However, substrate phosphorylation stimulated by 8Br-cAMP was significantly blunted in ACMs cultured overnight without insulin (Fig 2). Likewise, ACMs isolated from adult Akita mice had significantly blunted response to 8Br-cAMP as compared to WT. Furthermore, insulin did not enhance 8Br-cAMP-stimulated PKA substrate phosphorylation in Akita ACMs. While we cannot completely rule out changes in β-adrenergic receptor content or activity, these results support that the defect in PKA signaling induced by the absence of insulin is downstream of receptors and cAMP production.

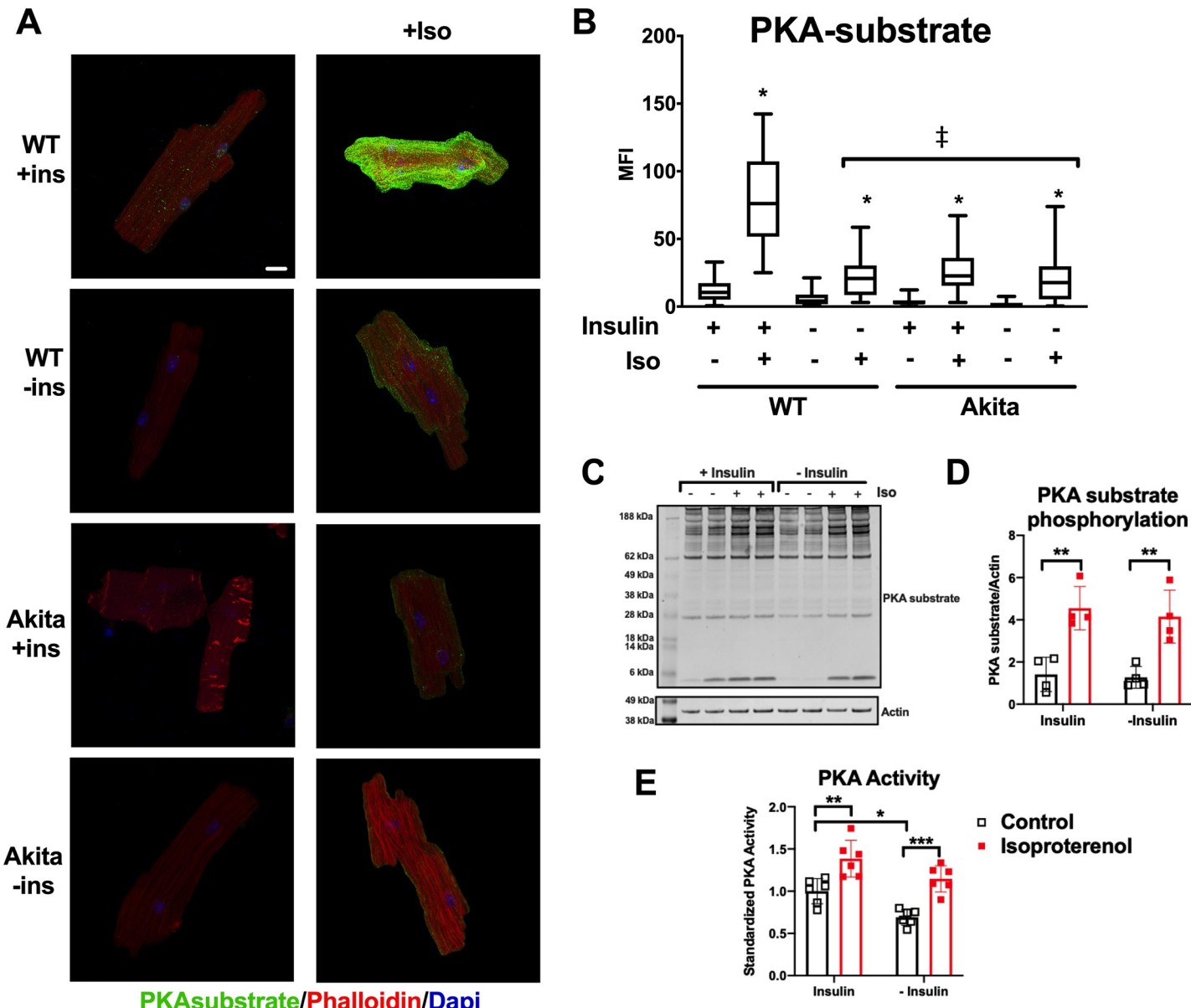

**Fig 1. Immunofluorescence detection of PKA signaling reveals a positive role of insulin.** (A) Adult mouse cardiomyocytes from wild-type or Akita mice were incubated overnight in the presence or absence of insulin (ins) and treated with 0.25μM isoproterenol for 30min as indicated. Cells were fixed and stained with rabbit anti-PKA substrate antibody visualized with Alexa 488 anti-rabbit secondary and with Alexa 568 labeled phalloidin. Maximum intensity micrographs were acquired as described in Materials and Methods and a representative image for each condition is shown. Scale bar 10μm. (B) Quantitation of mean fluorescence intensity (MFI) for PKA-substrate are presented as whisker plots that encompass data from at least 30 cells (n = 3 biological replicates, and at least 10 cells per experiment). The box dimensions extend from the $25^{th}$ to the $75^{th}$ percentiles; whiskers describe the minimum to maximum values. The median is plotted as a horizontal line within the box. *, ISO treatment caused a statistically significant increase in all conditions ($p < .001$) by one-way ANOVA. (‡) ISO stimulated samples from WT without insulin and Akita were significantly reduced ($p < .001$) compared to stimulated WT with insulin by one-way ANOVA with Tukey post hoc test. (C) Representative Western blot for anti-PKA substrate analysis of lysates from primary mouse cardiomyocytes treated as shown. (D) Quantitation of western blot experiments (n = 4). (E) PKA activity was measured in WT ACMs cultured in the presence or absence of insulin and with or without ISO stimulation as described in the Materials and Methods. PKA assays were performed in triplicate from 2 different ACM preparations. *, $p < .05$; **, $p < .005$ by two-way ANOVA with Tukey post hoc test.

Alterations in PKA signaling could be attributed to fluctuations in enzyme content. PKA catalytic subunit levels were therefore examined in ACMs from control and Akita mice cultured with or without insulin. As shown in S2 Fig, there were no significant differences in

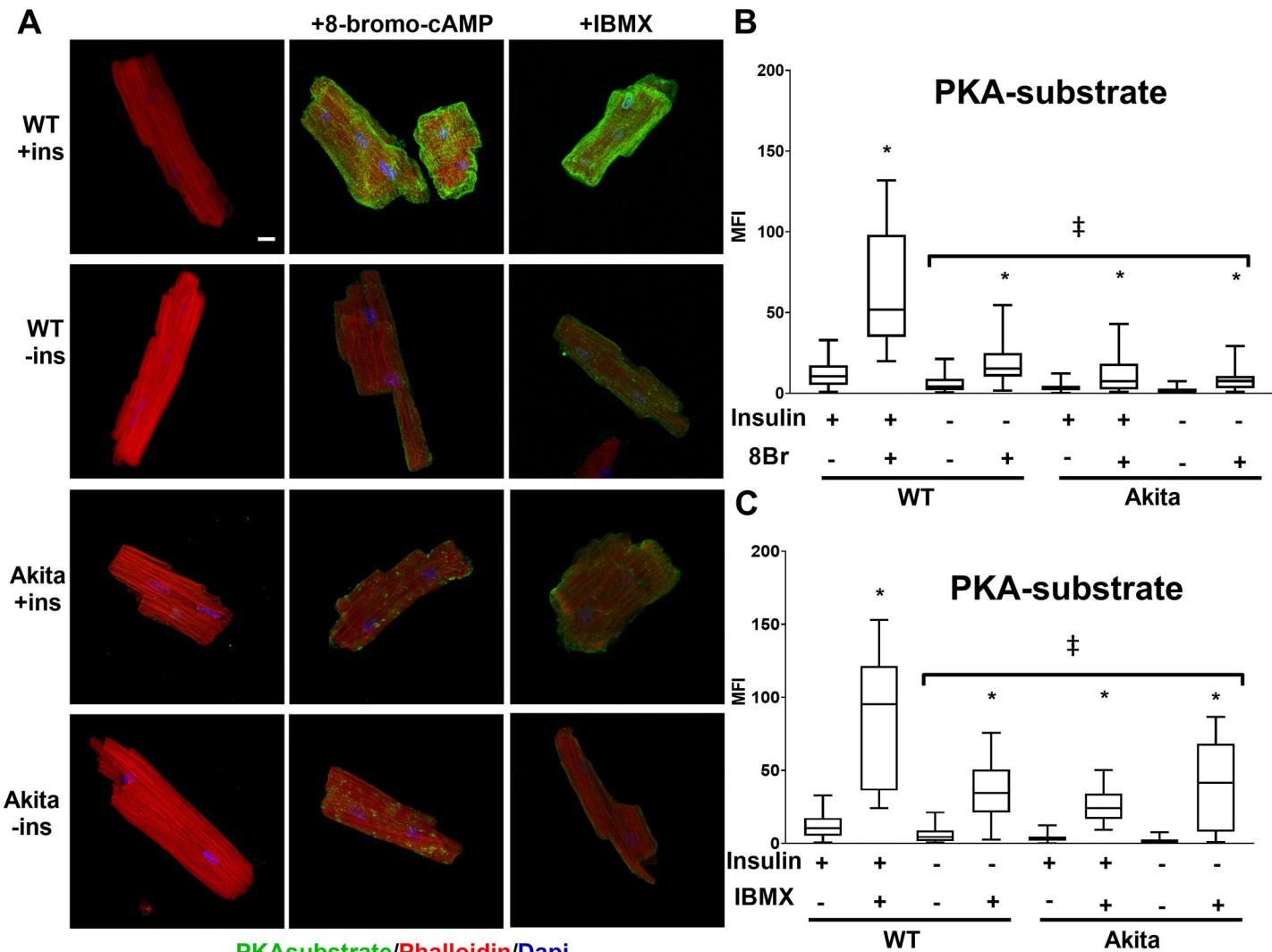

**Fig 2. The effect of insulin on PKA signaling occurs downstream of cAMP production.** (A) ACMs from wild-type or Akita mice were incubated overnight in the presence or absence of insulin (ins) and treated with 250µM 8-bromo-cAMP (8Br) or 500µM IBMX for 30min as indicated. Cells were fixed and stained with rabbit anti-PKA substrate antibody visualized with Alexa 488 anti-rabbit secondary and with Alexa 568 labeled phalloidin. Maximum intensity micrographs were acquired as described in Materials and Methods and a representative image for each condition is shown. Scale bar 10µm. (B&C) Quantitation of mean fluorescence intensity (MFI) for PKA-substrate are presented as whisker plots that encompass data from at least 18 cells, as detailed in Fig 1 (n = 3 biological replicates, and at least 6 cells per experiment). (*) Isoproterenol treatment caused a significant increase ($p < .001$) in all conditions. (‡) 8-bromo-cAMP or IBMX stimulated samples from WT without insulin and Akita were significantly reduced ($p < .001$) compared to stimulated WT with insulin. Statistics were performed by one-way ANOVA with Tukey post hoc test.

PKA content under the experimental conditions as determined by immunofluorescence or Western blot (not shown). This indicates the loss of PKA substrate phosphorylation is not mediated to decreased PKA catalytic subunit content.

## Phosphodiesterase inhibition increase PKA signaling but do not restore deficits induced by loss of insulin

Phosphodiesterases (PDEs) hydrolyze cAMP and are an essential component in modulating proper PKA signaling. Thus, insulin mediated changes in PDE activity could contribute to the observed effects on PKA signaling shown in Fig 1. We therefore examined the effect of

3-isobuty-1-methylxanthine (IBMX), a nonspecific phosphodiesterase inhibitor, on ACMs from control and Akita mice to determine whether blocking PDE activity is sufficient to recover PKA signaling. Addition of IBMX, in the absence of other PKA agonists, was sufficient to stimulate PKA signaling by 2.5-fold (Fig 2). ACMs isolated from adult Akita mice had significantly blunted response to IBMX as compared to WT. Furthermore, insulin did not enhance IBMX-stimulated PKA substrate phosphorylation in Akita ACMs. This demonstrates PDE inhibition is sufficient to stimulate PKA substrate phosphorylation similarly to a PKA agonist when insulin is present. However, this effect is blunted when WT ACMs are cultured in the absence of insulin. The stimulatory effects of IBMX are blunted in Akita ACMs and is unaffected by the insulin status.

We next sought to determine the combined effects of ISO and IBMX on PKA signaling. ACMs from control or Akita diabetic mice were cultured overnight in the presence or absence of insulin and then treated with combinations of IBMX and ISO. IBMX enhanced ISO stimulation of PKA substrate phosphorylation under all conditions (Fig 3). This demonstrates the importance of PDE activity in attenuating catecholamine mediated PKA signaling. However, the maximum PKA substrate phosphorylation was nevertheless blunted in control ACMs cultured in the absence of insulin. Akita ACMs exhibited a similar additive increase in PKA substrate phosphorylation with the combination of ISO and IBMX. However, the maximum intensity was decreased as compared to WT ACMs and unaffected by the insulin status.

Recent work has identified a relationship between cardiac insulin signaling and the content and activity of PDE4 [11]. Specifically, the hyperinsulinemia that occurs with type 2 diabetes is associated with increased PDE4B content which thereby decreases cAMP and attenuates β-adrenergic signaling [11]. We therefore tested to see if reciprocally the lack of insulin affects specific phosphodiesterases. Control ACMs were treated with a panel of phosphodiesterase inhibitors including IBMX (nonspecific PDE inhibitor), EHNA (PDE2 specific), MMPX (Calmodulin sensitive cyclic GMP specific), milrinone (PDE3 specific), clostramide (PDE3 specific), RO-20-1724 (PDE4 specific), rolipram (PDE4 specific), and zardavsine (PDE3/4 specific). All of the inhibitors, except EHNA, stimulated phosphorylation of PKA substrates. However, the most pronounced effect was observed with inhibitors of PDE4 (Fig 4). Next, we tested if PDE4 inhibition could increase PKA signaling in the absence of insulin. PKA signaling in ACMs from control mice treated with RO-20-1724 closely approximated the effects of IBMX. The drug enhanced PKA signaling on its own or in combination with ISO, but nevertheless signaling was decreased in ACMs cultured in the absence of insulin (S3 Fig). We also examined PDE4 by immunofluorescence and determined that the presence or absence of insulin had no effect on its content (S4 Fig). Consistent with previous reports [19, 20], this supports that PDE4 is the primary regulator of cAMP degradation in mouse cardiomyocytes, that its content is not affected by acute changes in insulin, and that its inhibition is not sufficient to fully recover PKA activity when insulin signaling is absent.

## Phosphorylation of PFK2 is decreased in diabetic conditions

Our results demonstrate that the absence of insulin affects PKA substrate phosphorylation downstream of β-adrenergic receptors and that this is not mediated by changes in adenylate cyclase or PDE activities. We next evaluated PKA signaling using phospho-specific antibodies to identify substrates that may be differentially phosphorylated in the absence of insulin. The bifunctional enzyme 6-phosphofructo-2-kinase/fructose-2,6-bisphosphatase (PFK-2) is a glycolytic regulator and substrate of PKA [21, 22]. When the cardiac isoform (gene produce of *pfkfb2*) is phosphorylated, PFK-2 increases the production of fructose-2,6-bisphosphate, a potent allosteric activator of the glycolytic enzyme phosphofructokinase-1 (PFK-1).

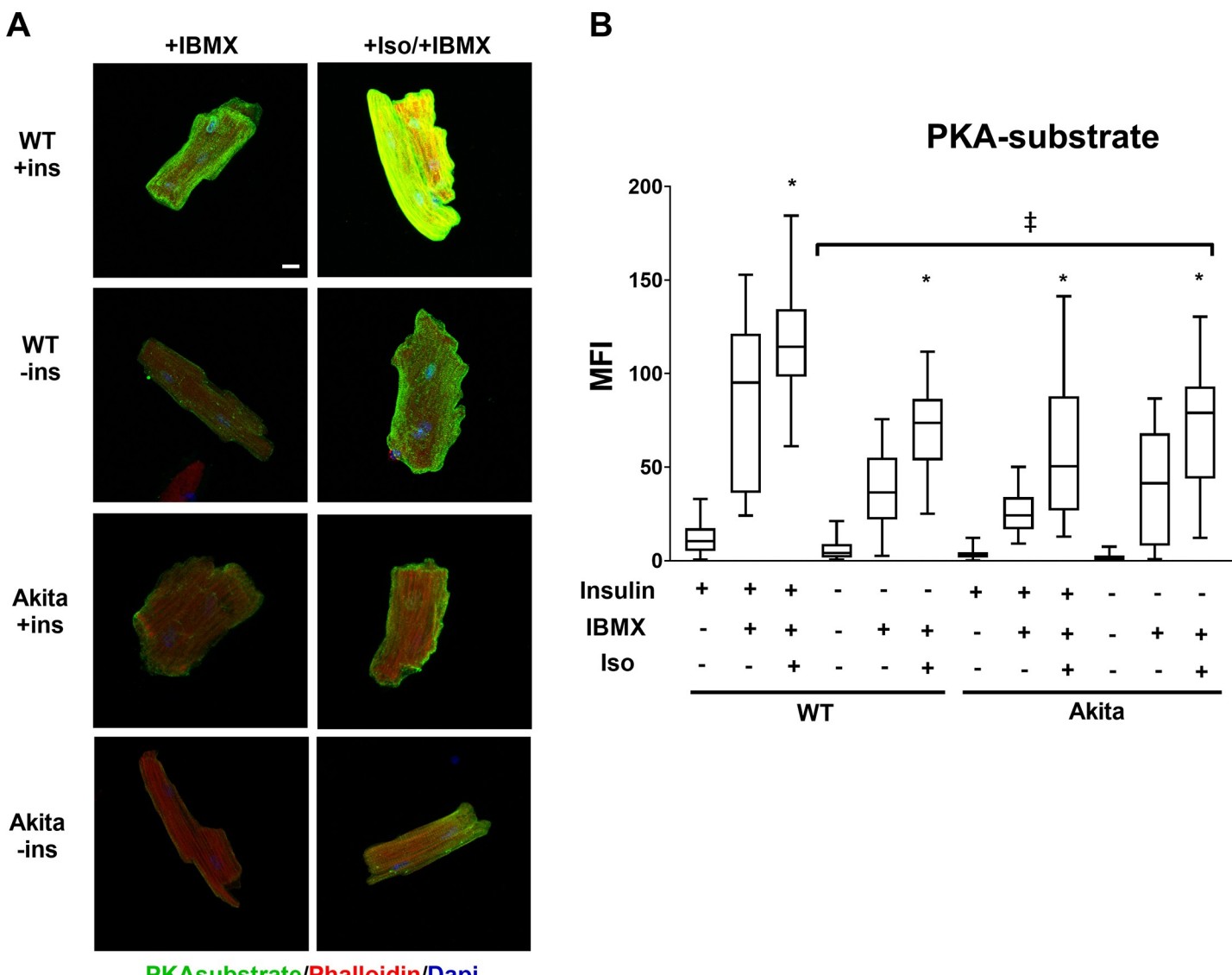

**PKAsubstrate/Phalloidin/Dapi**

**Fig 3. PKA signaling stimulated by the combination of isoproterenol and IBMX is decreased by the absence of insulin.** (A) ACMs from wild-type or Akita mice were incubated overnight in the presence or absence of insulin (ins) and treated with 500μM IBMX or .25μM Iso/500μM IBMX for 30min as indicated. Cells were fixed and stained with rabbit anti-PKA substrate antibody visualized with Alexa 488 anti-rabbit secondary and with Alexa 568 labeled phalloidin. Maximum intensity micrographs were acquired as described in Materials and Methods and a representative image for each condition is shown. Scale bar 10μm. (B) Quantitation of mean fluorescence intensity (MFI) for PKA-substrate are presented as whisker plots that encompass data from at least 18 cells, as detailed in Fig 1 (n = 3 biological replicates, and at least 6 cells per experiment). (*) Isoproterenol and IBMX treatment caused a significant increase ($p < .001$) compared to IBMX alone in all conditions. (‡) stimulated Akita and WT–ins conditions are significantly decreased compared to matching WT +ins conditions. Statistics were performed by one-way ANOVA with Tukey post hoc test.

Furthermore, we have previously reported that the content of PFK-2, is regulated by insulin signaling [4, 23]. Control ACMs were cultured for 18h in the presence or absence of insulin and then stimulated with either ISO or IBMX. As shown in Fig 5, PFK-2 phosphorylation was significantly increased in wildtype ACMs stimulated by either ISO or IBMX regardless of whether cells were cultured with insulin. However, for each condition the magnitude of PFK-2 phosphorylation was significantly less in ACMs cultured in the absence of insulin as compared to those cultured with insulin. In Akita ACMs, PFK-2 phosphorylation was significantly repressed basally and was largely unresponsive to stimulation by either ISO or IBMX.

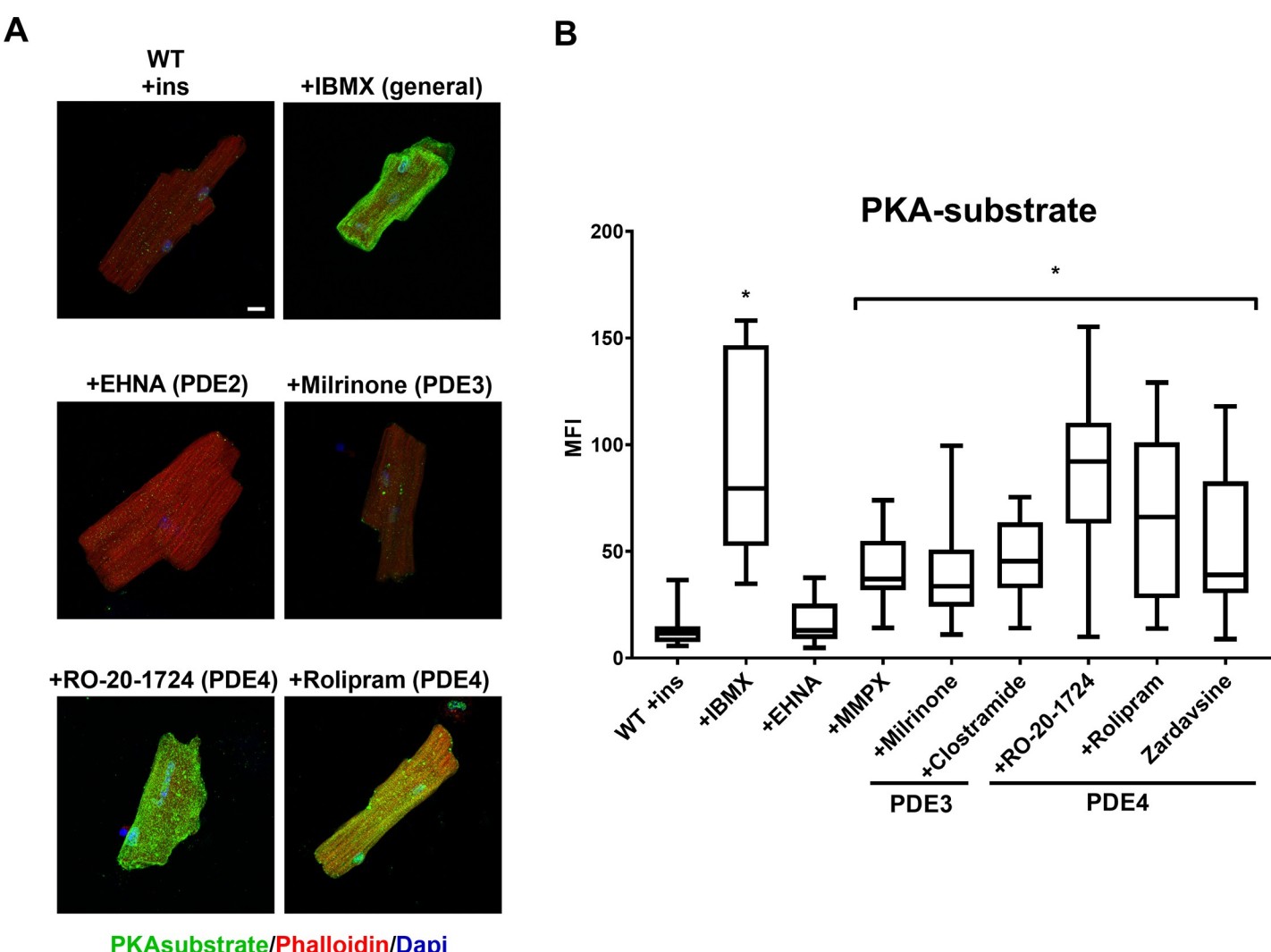

**Fig 4. Inhibition of PDE4 increases PKA signaling.** (A) ACMs from wild-type or Akita mice were incubated overnight in the presence insulin (ins) and treated with 500µM IBMX or indicated phosphodiesterase inhibitors. Cells were fixed and stained with rabbit anti-PKA substrate antibody visualized with Alexa 488 anti-rabbit secondary and with Alexa 568 labeled phalloidin. Maximum intensity micrographs were acquired as described in Materials and Methods and a representative image for each condition is shown. Scale bar 10µm. (B) Quantitation of mean fluorescence intensity (MFI) for PKA-substrate are presented as whisker plots that encompass data from at least 18 cells, as detailed in Fig 1 (n = 3 biological replicates, and at least 6 cells per experiment). *, $p < .001$ by one-way ANOVA with Tukey post hoc test.

We next examined whether the phosphorylation of other well-described PKA substrates exhibit insulin sensitivity. Phospholamban (PBN) is an inhibitor of the sarcoplasmic reticulum calcium dependent ATPase (SERCA2) channel. PBN mediated inhibition of SERCA2 is relieved upon phosphorylation by PKA, thereby increasing calcium cycling in response to β-adrenergic signaling [24]. Troponin I (TnI) is part of the calcium-sensitive troponin complex that decreases myosin-actin crossbridges. Phosphorylation of TnI by PKA decreases the sensitivity of the complex to calcium and is important for increasing the inotropic response [25, 26]. As shown in Fig 6, PBN and TnI were robustly phosphorylated in response to ISO or IBMX and this was unaffected by 18h of insulin starvation. These results demonstrate that there is a differential sensitivity to insulin among β-adrenergic targets, with the metabolic target, PFK-2, being significantly repressed while those involved in contractility are sustained.

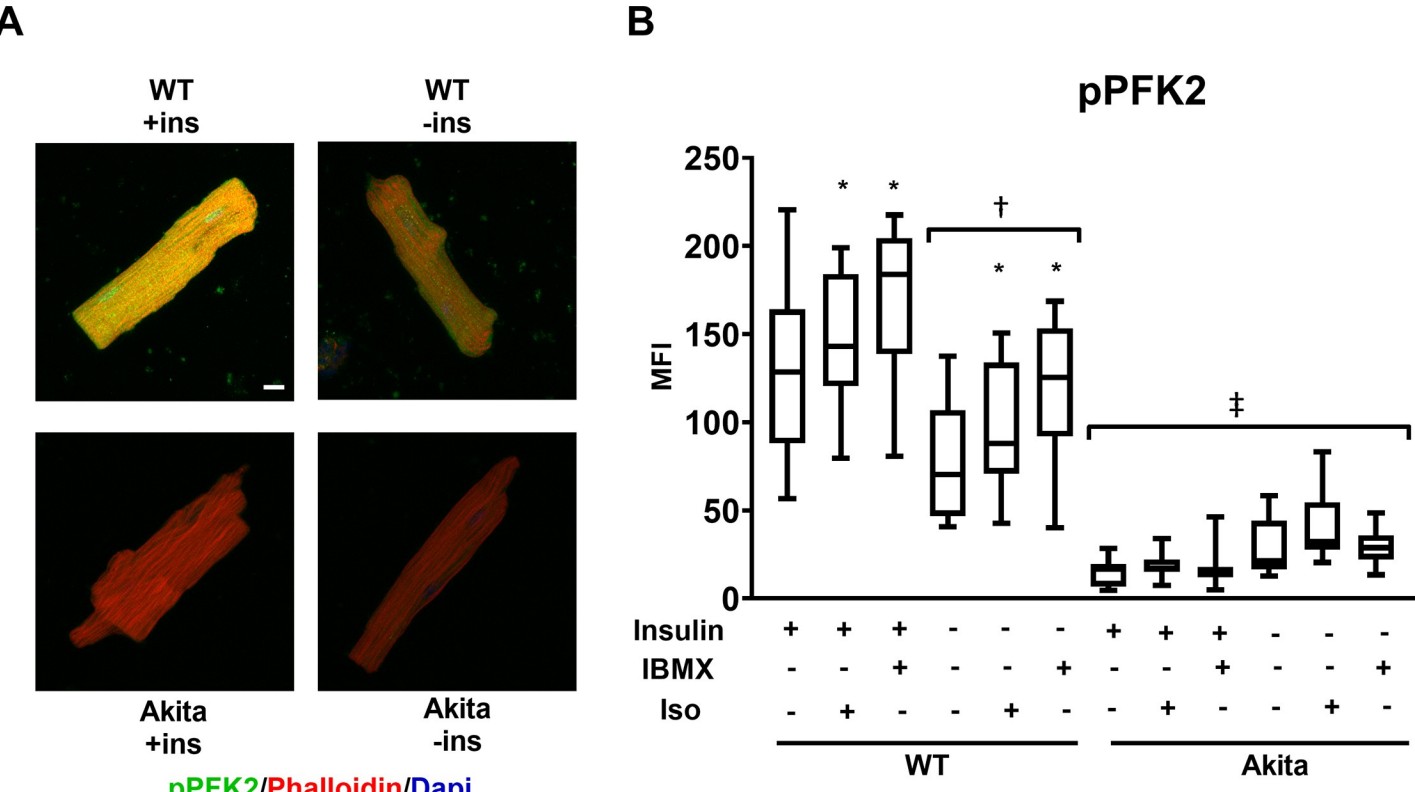

**Fig 5. Phosphorylation of PFK2 is decreased in cardiomyocytes cultured in the absence of insulin and with diabetes.** (A) ACMs from wild-type or Akita mice were incubated overnight in the presence or absence of insulin (ins) and treated with 500μM IBMX or .25μM Iso for 30min as indicated. Cells were fixed and stained with rabbit anti-phospo-PFK2 antibody visualized with Alexa 488 anti-rabbit secondary and with Alexa 568 labeled phalloidin. Maximum intensity micrographs were acquired as described in Materials and Methods and a representative image for select conditions are shown. Scale bar 10μm. (B) Quantitation of mean fluorescence intensity (MFI) for pPFK2 are presented as whisker plots that encompass data from at least 18 cells, as detailed in Fig 1 (n = 3 biological replicates, and at least 6 cells per experiment). (*) IBMX or Isoproterenol treatment causes a significant increase ($p < .001$) compared to untreated WT cells; (†) Each WT (-ins) condition is significantly decreased ($p < .001$) compared to WT (+ins); (‡) all Akita conditions are significantly decreased ($p < .001$) compared to comparable WT conditions. Statistics performed by one-way ANOVA with Tukey post hoc test.

## PKA substrate phosphorylation remains depressed upon acute insulin administration

Lastly, we sought to determine whether the loss of insulin has a sustained effect on PKA signaling. ACMs from wildtype mice were cultured for 18h in the absence of insulin and then stimulated acutely with combinations of ISO and insulin. As shown in Fig 7A and 7B, the acute addition of insulin alone failed to increase PKA substrate phosphorylation. Furthermore, the acute addition of insulin had no additive effect on ISO mediated PKA substrate phosphorylation. This lack of effect was not from deficiencies in the insulin signaling pathway. A time course of Akt phosphorylation was examined by Western blot in ACMs cultured in the presence or absence of insulin and then acutely treated with combinations of insulin and ISO (Fig 7C and 7D). ACMs that had been cultured overnight with insulin had low levels of Akt phosphorylation. Acute additions of ISO and insulin minimally increased Akt phosphorylation. This suggests that desensitization of the pathway occurs when cells are cultured continuously with insulin. In contrast, ACMs cultured overnight in the absence of insulin showed a robust increase in Akt phosphorylation upon acute insulin treatment regardless of the presence of ISO. The signal was maximal at 10min and then decreased over time. Thus, the lack of acute

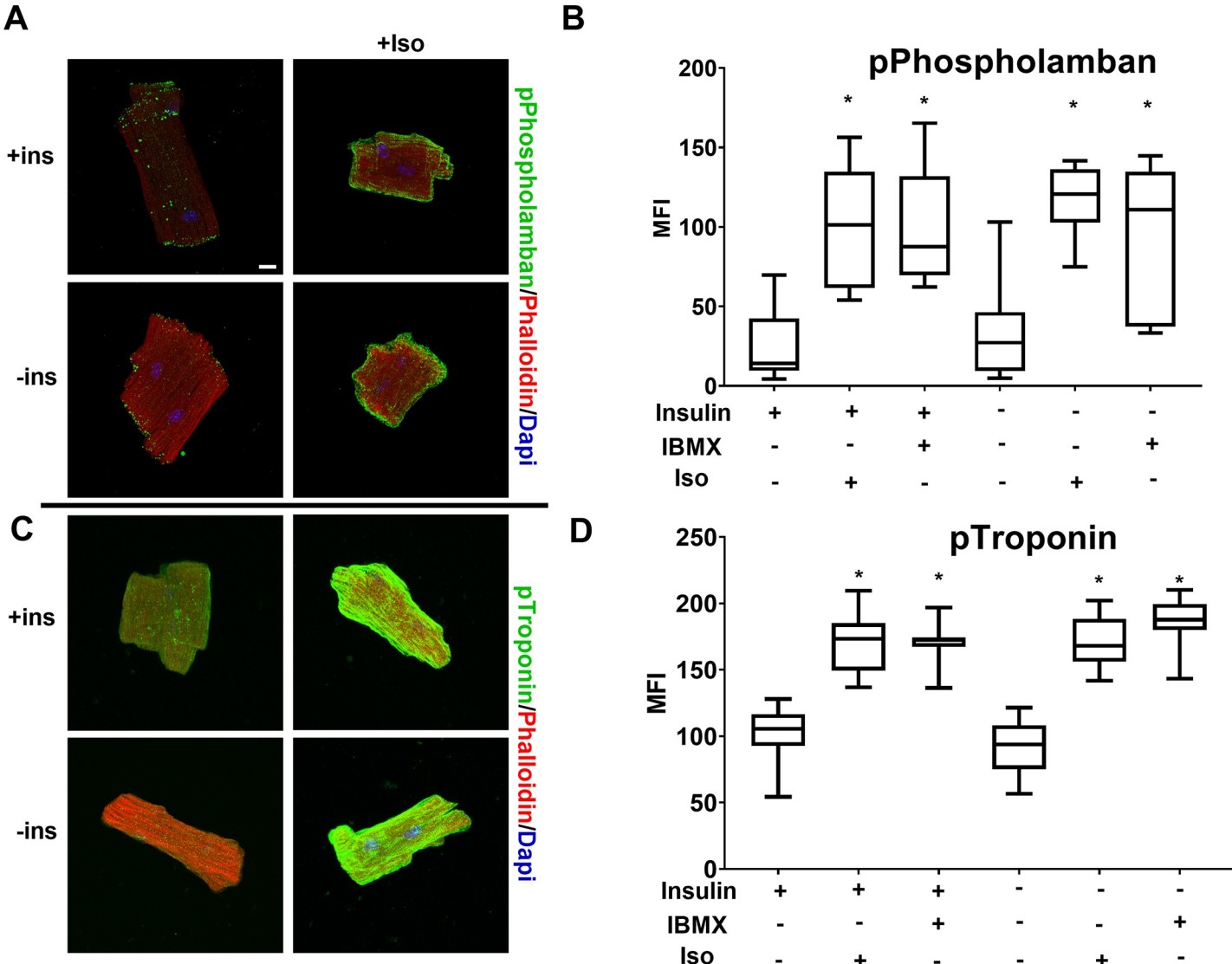

**Fig 6. Phosphorylation of β-adrenergic targets Phospholamban and Troponin are unchanged under diabetic conditions.** (A&C) Primary mouse cardiomyocytes from wild-type mice were incubated overnight in the presence or absence of insulin (ins) and treated with .25μM Iso or 500μM IBMX for 30min as indicated. Cells were fixed and stained with rabbit anti-phospho-phospholamban (A) or rabbit anti-phospho-Troponin (C) antibody visualized with Alexa 488 anti-rabbit secondary and with Alexa 568 labeled phalloidin. Maximum intensity micrographs were acquired as described in Materials and Methods and a representative image for select conditions are shown. Scale bar 10μm. (B&D) Quantitation of mean fluorescence intensity (MFI) for pPhospholamban (B) or pTroponin (D) are presented as whisker plots that encompass data from at least 18 cells, as detailed in Fig 1 (n = 3 biological replicates, and at least 6 cells per experiment). (*) IBMX or isoproterenol treatment causes a significant ($p < .001$) increase by one-way ANOVA with Tukey post hoc test.

effect by insulin on PKA signaling was not due to unresponsiveness of the insulin signaling pathway.

## Discussion

β-Adrenergic and insulin signaling pathways are the primary means of modulating moment-to-moment changes in cardiac function and metabolic substrate selection. Nevertheless, interactions between these two pathways are not fully understood. This is important to understand in regard to diabetes where insulin signaling is disrupted and PKA signaling is dysfunctional

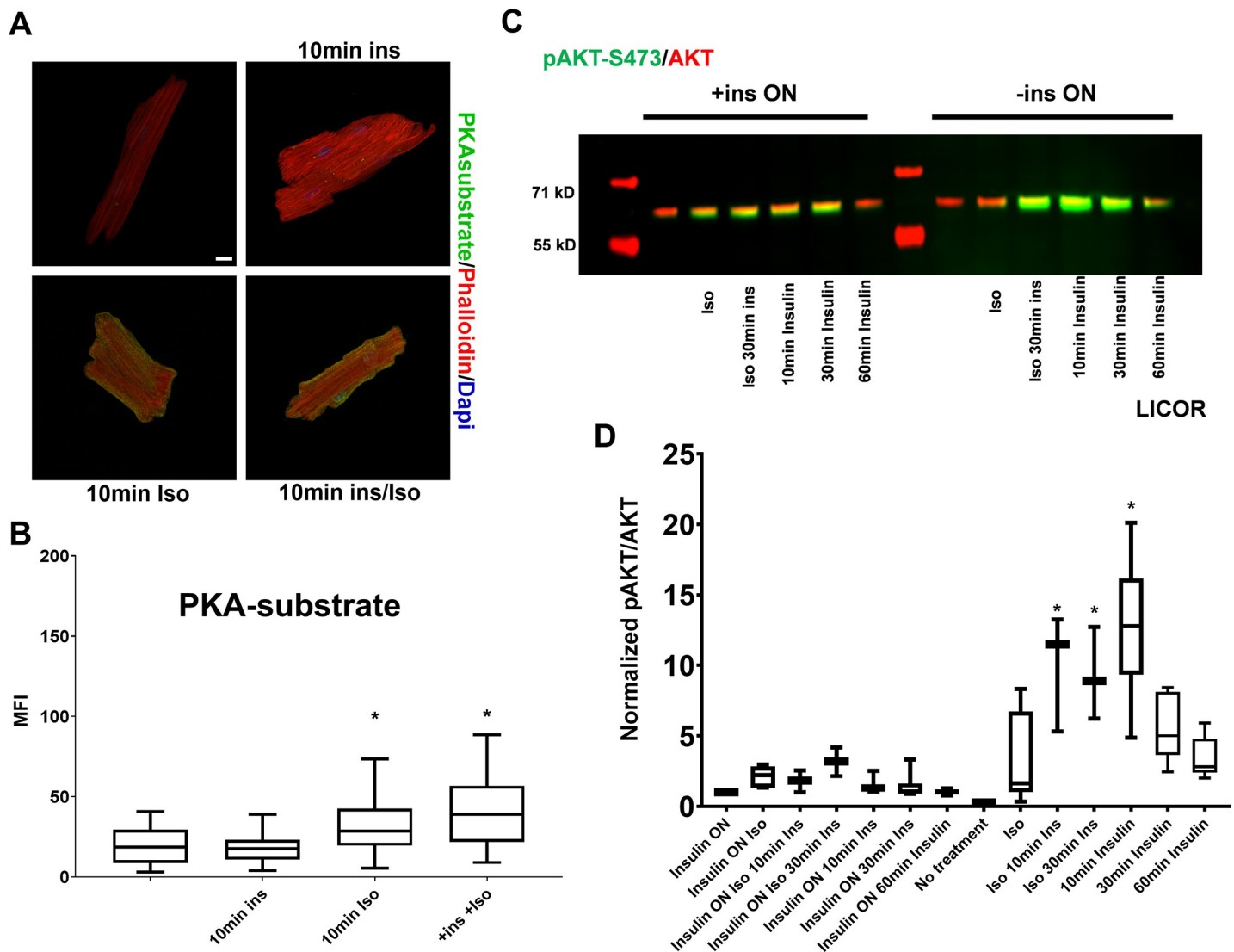

**Fig 7. Short term insulin stimulation does not rescue PKA-substrate phosphorylation.** (A) Primary mouse cardiomyocytes from wild-type were incubated overnight in the absence of insulin and treated with 0.25μM Isoproterenol or insulin for 10min as indicated. Cells were fixed and stained with rabbit anti-PKA substrate antibody visualized with Alexa 488 anti-rabbit secondary and with Alexa 568 labeled phalloidin. Maximum intensity micrographs were acquired as described in Materials and Methods and a representative image for each condition is shown. Scale bar 10μm. (B) Quantitation of mean fluorescence intensity (MFI) for PKA-substrate are presented as whisker plots that encompass data from at least 18 cells, as detailed in Fig 1 (n = 3 biological replicates, and at least 6 cells per experiment). (*) Isoproterenol causes a significant increase ($p < .05$) by one-way ANOVA. (C) Representative western blot of anti-phospho-Akt-S473 (green) and total Akt (red) analysis of lysates from primary mouse cardiomyocytes treated as in B. (D) Western blot quantitation presented as whisker plots as described in Fig 1. (*) short term insulin stimulation causes a statistically significant increase in AKT phosphorylation at S473 by one-way ANOVA with Tukey post hoc test.

[3]. In an effort to further understand the interrelationship of these two pathways, we used adult mouse cardiomyocytes as a model system. ACMs have the advantage that they more closely represent in vivo signaling and metabolic conditions as compared to immortalized cardiomyoblasts, such as H9c2 and HL-2 cells [27]. ACMs can also be isolated from different genetic and disease models, as in the Akita mice used here, to interrogate alterations in function at the cellular level. A disadvantage is that mouse ACMs are not amenable to genetic manipulation and have a limited lifespan. For these reasons, we chose to optimize conditions to monitor PKA signaling using an immunofluorescence technique. This approach, to the best of our knowledge, has not been taken before in adult ACMs.

The antibody that we used to measure PKA signaling is validated by the observations that the immunofluorescence intensity is responsive to a β-agonist (ISO), a PKA agonist (8Br-cAMP), and PDE inhibitors. Thus, by this methodology we were able to visualize how insulin affects PKA signaling at the cellular level, specifically in a homogenous cardiomyocyte population, and independently of other systemic factors. Little difference was seen in the subcellular distribution and pattern of PKA substrate phosphorylation in wildtype cells stimulated with either ISO or 8Br-cAMP when cells were culture in the presence of insulin. This suggests that similar pools of PKA were activated by both agonists, resulting in phosphorylation of downstream substrates throughout the cell. When wildtype ACMs were cultured in the absence of insulin, though, PKA phosphorylation was substantially blunted in response to all three agonists examined. In Akita ACMs PKA substrate phosphorylation was decreased as compared to wildtypes. Furthermore, insulin had no beneficial effects on PKA substrate phosphorylation in Akita ACMs. Interestingly, though, when PKA phosphorylation was examined by Western blot analysis using the same antibody as in the immunofluorescence experiments, no significant differences were seen in the pattern of proteins phosphorylated. This was further corroborated by measuring total PKA activity (Fig 1E). We saw that the absence of insulin reduced basal PKA activity. However, ISO stimulated PKA activity similarly in ACMs cultured with or without insulin. The cause of this reduced basal PKA activity is not due to reduced PKA catalytic subunit content (S2 Fig) and needs further investigation.

One possibility is that the absence of insulin induces changes in ACM morphology and impairs PKA substrate detection by immunofluorescence. This is countered, though, by the finding that the phosphorylation of PBN and TnI were robustly induced by agonists even in the absence of insulin. Rather, we suggest that there is a differential recognition of PKA substrate epitopes depending upon the protein status. The immunofluorescence protocol maintains proteins in a native conformation, as compared to Western blot where proteins are denatured.

The effects of insulin on global PKA substrate phosphorylation followed a similar pattern to that observed with phospho-PFK-2 staining. The cardiac isoform of PFK-2 is phosphorylated in response to insulin or β-agonists to increase the levels of fructose-2,6-bisphosphate, an allosteric activator of PFK-1 [28]. This serves to increase glycolytic flux. We have previously reported that PFK-2 content is decreased in the absence of insulin [3]. In the streptozotocin toxin-induced type 1 diabetic model this results in a constitutive decrease in its content. In addition, the PFK-2 that remains is not phosphorylated in response to PKA activation. Our immunofluorescence data follows a similar pattern. The absence of insulin decreases basal PFK-2 phosphorylation and dampens its phosphorylation in response to PKA agonists (Fig 5). In Akita ACMs, we observed a constitutive decrease in phospho-PFK-2. Interestingly, though, culturing Akita ACMs overnight with insulin failed to rescue PFK-2 phosphorylation. This suggests that the mechanisms that normally regulate the expression and activation of PFK-2 are not affected by administration of insulin for 18h. While the pattern of PKA substrate phosphorylation approximates that of phospho-PFK-2, we cannot rule out that other PKA substrates are also affected by the insulin status. Furthermore, insulin also activates the mitogen-activated protein kinase/extracellular-regulated kinase pathway and effects related to the stimulation of these other signaling cascades must be further evaluated [9].

We demonstrate here that insulin affects PKA signaling in ACMs. Reciprocally, previous studies have shown that PKA can affect insulin signaling. In neonatal rat cardiomyocytes and mice with chronic β-adrenergic stimulation there is a PKA-dependent decrease in insulin signaling [29, 30]. The mechanism involves insulin receptor desensitization, mediated by Akt [30], and is manifested by a decrease in GLUT4 content and translocation upon insulin stimulation [29]. This PKA-mediated effect contributes to the insulin resistance that is manifested in

failing hearts. These role of PKA in modulating insulin signaling, though, are dependent upon the duration of PKA activation. With short-term activation of PKA there is synergistic enhancement of Akt phosphorylation, GLUT4 translocation, and glucose uptake [30, 31]. This mediates the increase in glucose uptake and oxidation in response to acute β-adrenergic stimulation.

Our results provide new insights into how diabetes may impact the heart. We show a decrease in PKA signaling in adult cardiomyocytes when insulin is absent. A novelty of this study was the use of immunofluorescence microscopy as a means of monitoring PKA signaling. As we show, other methodologies, such as Western blot, would have missed these apparent changes in substrate phosphorylation. Another novel aspect of this study was using primary adult cardiomyocytes isolated from Akita mice. Immortalized cells cannot recapitulate the unique morphology and metabolic aspects of primary cells. We demonstrate that the lack of insulin affects the phosphorylation of PFK-2, while the phosphorylation of contractile proteins was similar in control and Akita ACMs. Future studies must be performed to more exhaustively identify what other substrates may be affected.

## Supporting information

**S1 Fig. β-adrenergic stimulation increases PKA substrate phosphorylation in adult mouse cardiomyocytes.** (A) ACMs from wild-type mice were incubated overnight in the presence or absence of insulin (ins) and treated with 0.25μM Isoproterenol for 1, 5, and 30min as indicated. Cells were fixed and stained with rabbit anti-PKA substrate antibody visualized with Alexa 488 anti-rabbit secondary and with Alexa 568 labeled phalloidin. Maximum intensity micrographs were acquired as described in Materials and Methods and a representative image for each condition is shown. Scale bar 10μm. (B) Quantitation of mean fluorescence intensity (MFI) for PKA-substrate are presented as whisker plots that encompass data from at least 18 cells, as detailed in Fig 1 (n = 3 biological replicates, and at least 6 cells per experiment). *, significant difference ($p < .001$) by one-way ANOVA with Tukey post hoc test.
(TIF)

**S2 Fig. PKA catalytic subunit levels are unchanged under diabetic conditions.** (A) ACMs from wild-type or Akita mice were incubated overnight in the presence or absence of insulin (ins) and treated with 0.25μM Isoproterenol or 250μM 8-bromo-cAMP for 30min as indicated. Cells were fixed and stained with rabbit anti-PKA catalytic subunit antibody visualized with Alexa 488 anti-rabbit secondary and with Alexa 568 labeled phalloidin. Maximum intensity micrographs were acquired as described in Materials and Methods and a representative image for each condition is shown. Scale bar 10μm. (B) Quantitation of mean fluorescence intensity (MFI) for PKA-substrate are presented as whisker plots that encompass data from at least 18 cells, as detailed in Fig 1 (n = 3 biological replicates, and at least 6 cells per experiment).
(TIF)

**S3 Fig. PDE4 inhibition increases PKA signaling.** (A) ACMs from wild-type mice were incubated overnight in the presence or absence of insulin (ins) and treated with 0.25μM Isoproterenol and/or 10μM RO. Cells were fixed and stained with rabbit anti-PKA substrate antibody visualized with Alexa 488 anti-rabbit secondary and with Alexa 568 labeled phalloidin. Maximum intensity micrographs were acquired as described in Materials and Methods and a representative image for each condition is shown. Scale bar 10μm. (B) Quantitation of mean fluorescence intensity (MFI) for PKA-substrate are presented as whisker plots that encompass data from at least 18 cells, as detailed in Fig 1 (n = 3 biological replicates, and at least 6 cells per experiment). *, significant difference ($p < .001$) by one-way ANOVA with multiple

comparisons using Tukey's test.
(TIF)

**S4 Fig. PDE4D protein levels are unchanged in diabetic or β-adrenergic stimulation conditions.** (A) Primary mouse cardiomyocytes from wild-type (C57-B6) or Akita were incubated overnight in the presence or absence of insulin (ins) and treated with 0.25μM Isoproterenol or 500μM IBMX for 30min. Cells were fixed and stained with rabbit anti-PDE4D antibody visualized with Alexa 488 anti-rabbit secondary and with Alexa 568 labeled phalloidin. Maximum intensity micrographs were acquired as described in Materials and Methods and a representative image for each condition is shown. Scale bar 10μm. (B) Quantitation of mean fluorescence intensity (MFI) for PDE4D are presented as whisker plots that encompass data from at least 18 cells, as detailed in Fig 1 (n = 3 biological replicates, and at least 6 cells per experiment). *, significant difference ($p < .001$) by one-way ANOVA with multiple comparisons using Tukey's test.
(TIF)

**S1 Raw images.**
(PDF)

## Author Contributions

**Conceptualization:** Craig A. Eyster, Kenneth M. Humphries.

**Investigation:** Craig A. Eyster, Satoshi Matsuzaki, Maria F. Newhardt, Jennifer R. Giorgione.

**Methodology:** Craig A. Eyster, Maria F. Newhardt.

**Supervision:** Kenneth M. Humphries.

**Writing – original draft:** Kenneth M. Humphries.

**Writing – review & editing:** Craig A. Eyster, Kenneth M. Humphries.

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
