## [Decision Letter · Decision Letter 0]

8 May 2020

PONE-D-20-09116

Diabetes induced decreases in PKA signaling in cardiomyocytes: the role of insulin

PLOS ONE

Dear Dr. Humphries,

Thank you for submitting your manuscript to PLOS ONE. After careful consideration, we feel that it has merit but does not fully meet PLOS ONE’s publication criteria as it currently stands. Therefore, we invite you to submit a revised version of the manuscript that addresses the points raised during the review process.

Your manuscript was reviewed by two knowledgeable referees in this area and their comments are appended. As you will see they had a numerous major concerns that will need to be properly addressed by the authors before I can proceed further. In particular, they both raised technical issues including the fluorescent-based evaluation methods. In addition, they felt that conclusion drawn is not fully supported by the results presented. The authors need to address/respond all their comments to fully satisfy both reviewers.

We would appreciate receiving your revised manuscript by Jun 22 2020 11:59PM. To enhance the reproducibility of your results, we recommend that if applicable you deposit your laboratory protocols in protocols.io, where a protocol can be assigned its own identifier (DOI) such that it can be cited independently in the future. For instructions see: http://journals.plos.org/plosone/s/submission-guidelines#loc-laboratory-protocols

We look forward to receiving your revised manuscript.

Kind regards,

Makoto Kanzaki, Ph.D.

Academic Editor

PLOS ONE

Journal Requirements:

Reviewers' comments:

Reviewer's Responses to Questions

**Comments to the Author**

1. Is the manuscript technically sound, and do the data support the conclusions?

Reviewer #1: Partly

Reviewer #2: Partly

2. Has the statistical analysis been performed appropriately and rigorously? 

Reviewer #1: Yes

Reviewer #2: I Don't Know

3. Have the authors made all data underlying the findings in their manuscript fully available?

Reviewer #1: Yes

Reviewer #2: Yes

4. Is the manuscript presented in an intelligible fashion and written in standard English?

Reviewer #1: Yes

Reviewer #2: No

5. Review Comments to the Author

Reviewer #1: This study shows that PFK-2, a regulator of glucose metabolism, of wild type cardiac myocytes is downregulated in insulin-dependent manner. However, PFK-2 of Akita diabetic model mice was decreased and was not recovered by the addition of insulin. It was suggested that insulin-dependent decrease of PFK-2 expression is a mechanism of metabolic regulation by insulin in diabetes. However, there are several concerns as described below.

However, in vitro assay system does not mimic in vivo phenomena as described in following comments. It is difficult to conclude that PFK-2 is an insulin-dependent target molecule in diabetic heart. The followings are comments.

1. When the effects of insulin are due to the decrease of PFK-2 expression level, overexpression of PFK-2 should restore the effects of cAMP signal-induced responses. Authors described at lines 398 o 399 that ‘a disadvantage is that ACMs are not amenable to genetic manipulation and have a limited lifespan’. However, there is the report that adenovirus is successfully used for expression of FRET biosensors in adult mouse cardiomyocytes (Methods Mol Biol. 2015; 1294:103-15). Therefore, expression of genes of interest is possible. The experiment of PFK-2 overexpression should be done.

2. In Fig. 1B, PKA substrates detected by anti-PKA consensus sequence-antibody represents all the proteins phosphorylated by PKA. However, minor but important PKA-phosphorylating proteins are not apparent from this detection system.

3. Immunofluorescence of PKA substrates is not quantitative. The mRNA expression levels should be measured. Then, it will provide information that insulin regulates transcription, translation, stability of protein.

4. In many cases, signaling intensity by Western blot is shown as a ratio of phosphorylated (active) form to total protein content. When the phosphorylated form decreases in diseased state and the total protein content also decreases, the ratio of phosphorylated form to total protein content is almost the same as that of control group. In this case, fold activation of diseased mice by β-adrenergic receptor stimulation will be about the same as control mice. When fold stimulation is same between control and diseased states, it cannot be concluded that decreased PKA signaling is due to the decreased expression of total protein content.

5. PKA signaling is mediated by signaling complex being comprised of PKA, A-kinase anchoring protein (AKAP), target protein, and other signaling proteins. Intensity, efficiency and specificity of PKA signaling depend on AKAPs in many cases. Therefore, the effects of insulin on decreased PKA signaling may be explained by the decreased expression of AKAPs. Western blot and immunofluorescence data of AKAPs expressing in the myocytes are necessary to substantiate the conclusion.

6. When phosphorylation states is really wanted to be examined, phospho-proteomics analysis may be better than Western blot instead of using anti-PKA consensus sequence-antibody.

7. Insulin has growth factor-like effects. Therefore, overnight culture without insulin may affect cell growth or maintenance. Thus, the effects of insulin on the isolated cells cultured overnight without insulin may be growth factor-like actions but not metabolic actions as suggested.

8. Insulin treatment may modify epigenetic modification during development of diabetes. It is interesting to examine insulin-induced epigenetic modification in myocytes of Akita diabetic model mice.

Minor comment

1. At lines 456 to 458, authors described that ‘our results support that the deficiency of PKA signaling is not mediated by loss of β-adrenergic receptors, PKA protein, or cAMP production/degradation’. However, authors do not measure number of β-adrenergic receptors, and amounts of PKA protein and PDEs/adenylyl cyclases. It is too much to say that.

2. There are many proteins that their activities re regulated by insulin in myocytes. The decrease of PFK-2 level may not be enough to explain insulin-dependent metabolic changes in the heart.

Reviewer #2: In the manuscript titled “Diabetes induced decreases in PKA signaling in cardiomyocytes: the role of insulin”, the authors test the hypothesis that decreased insulin signaling contributes to dysfunctional PKA response in adult rat ventricular myocytes. The authors use a novel technique of immunofluorescence microscopy to monitor PKA signaling in fixed myocytes cultured from WT or Akita type 1 diabetic mice. The results demonstrate that a lack of insulin in culture conditions is associated with reduced PKA signaling when measured by immunofluorescence. Further, the authors explore the possibility that phosphorylation of the glycolytic regulator PFK2 and not contractile proteins were responsible for the overall reduction in PKA signaling. The authors conclude that deficient insulin signaling decreases PKA signaling in adult myocytes, which may provide insights into how diabetes impacts the heart.

Major issues

1. There are several inconsistencies between the written text and what is shown in the figures. For example, line 191 onward describes WB data in Figure 1B, Figure 1B however, shows something different, not WB data. Continuing from that point, the authors state in line 199 "culturing myocytes in the presence of insulin for 18hrs had no enhancing effect on PKA substrate staining upon ISO stimulation". The authors need to be clear if they are comparing this to WT or baseline, as compared to baseline there is an effect with ISO stimulation in all groups. Far too much of my time was spent trying to interpret the data and I would urge the authors to proofread and ensure the manuscript is concise and to the point.

2. No in vivo mouse data is provided in the manuscript. While the Akita model has previously been reported to be associated with diabetes, there is no data provided demonstrating that this model serves as a mechanism of low insulin as a result of diabetes in this study. It would be interesting to determine the overall insulin levels of Akita mice and how does this compare to WT without insulin.

3. The use of immunofluorescence microscopy to monitor PKA signalling, as has been demonstrated in this study, has not been done before in adult myocytes. I am, therefore, surprised the authors have not validated this technique with other well-know measurements of PKA signaling such as a PKA activity kit to confirm the results. Immunofluorescence can often lead to false results; therefore the use of an antibody control is strongly advised. The manuscript would also benefit from clarification of how the mean fluorescence intensity images were normalized and the control measures taken to ensure there are no false positive signals.

4. Extended time in culture leads to great changes in myocyte morphology. For example, the structural integrity of the cells, such as t-tubule organization becomes disrupted after prolonged culture. Both insulin receptors and B-receptors are located on the t-tubule membrane, thus it cannot be ruled out that signaling alterations could be a consequence of culture. Moreover, just the presence of insulin in the culture media may have metabolic benefits that prove more advantageous for long culture conditions. To address this, a time-course study would be useful, starting with the effects of insulin on PKA signaling in freshly isolated cells.

5. There are concerns regarding the authors’ interpretation of the pPFK2 data. Especially, with the conclusion that PFK2 phosphorylation was decreased in an insulin-dependent manner. Although blunted compared with WT insulin hearts, WT hearts in the absence of insulin showed increased pPFK2 in response to ISO and IBMX. In the Akita hearts, however, ISO and IBMX failed to have any effect. This suggests that it is not just the absence of insulin, but something specific to the Akita mice leading to decreased pPFK2 activity.

Minor issues

1. Were the stats performed on cells or animals, have the authors considered performing nested statistics to account for cells and animals?

2. It would be useful if the graphs displayed individual data points.

3. The authors state PKA staining is found on the z-bands, but there is no mention as to which software was used to determine the position on the PKA activity staining?

6. PLOS authors have the option to publish the peer review history of their article (what does this mean?). If published, this will include your full peer review and any attached files.

Reviewer #1: No

Reviewer #2: No

---

## [Author Response · Author response to Decision Letter 0]

30 Jun 2020

We’d like to thank the reviewers for their careful evaluation of our manuscript. We have addressed their concerns to the best of our ability. This includes extensive revision of the manuscript and an additional experiment. 

Reviewer 1:

Reviewer 1’s general comment is that “It is difficult to conclude that PFK-2 is an insulin-dependent target molecule in diabetic heart.” However, this current study is supported by, and builds upon, our previous report that PFK-2 content is regulated by insulin signaling (see Bockus et al. 2017, Journal of American Heart Association; https://www.ahajournals.org/doi/full/10.1161/jaha.117.007159). We concur that other targets are likely to be affected by the presence or absence of insulin, but this and our previous studies support phosphorylation of PFK-2 is abnormal in the diabetic heart.

1. “When the effects of insulin are due to the decrease of PFK-2 expression level, overexpression of PFK-2 should restore the effects of cAMP signal-induced responses. Authors described at lines 398 o 399 that ‘a disadvantage is that ACMs are not amenable to genetic manipulation and have a limited lifespan’. However, there is the report that adenovirus is successfully used for expression of FRET biosensors in adult mouse cardiomyocytes (Methods Mol Biol. 2015; 1294:103-15). Therefore, expression of genes of interest is possible. The experiment of PFK-2 overexpression should be done.”

Thank you for providing the suggestion and the reference. As noted in this elegant paper by Zaccolo et al., there are limitations with adenovirus expression in adult mouse cardiomyocytes. First, it takes a high MOI for protein expression. We have also found this to be true, and as Zaccolo noted, this has toxicity issues. Secondly, the referenced paper reports that the timing of an adenoviral transduction experiment is dependent upon the protein that is being expressed. In the case of the referenced work, they are expressing a FRET reporter with favorable fluorescent properties that allows detection by 24h. In our experience it takes approximately 48-72h, if at all, to detect the expression of unlabeled proteins in adult mouse cardiomyocytes. Unfortunately, at this point there is also significant loss of viable cells which may be further confounded by the insulin status. Thus, while such a rescue experiment would help to support our conclusions it is technically unfeasible. We have edited the manuscript to reflect some of the limitations. Furthermore, we focus on the phosphorylation status of PFK-2 and not on its total content. 

2. In Fig. 1B, PKA substrates detected by anti-PKA consensus sequence-antibody represents all the proteins phosphorylated by PKA. However, minor but important PKA-phosphorylating proteins are not apparent from this detection system. 

We agree that there might be additional substrates we can’t detect by Western. We clarified in the text that other substrates may be phosphorylated but not detected (lines 220-222). 

3. “Immunofluorescence of PKA substrates is not quantitative. The mRNA expression levels should be measured. Then, it will provide information that insulin regulates transcription, translation, stability of protein.”

We have attempted to make the immunofluorescence data as quantitative as possible by keeping the experimental conditions constant. This includes how the cardiomyocytes were prepared and the microscopy conditions. 

We have previously shown that in the diabetic heart PFK-2 protein levels decrease without a change in transcript levels (Bockus et al., 2017). We concluded that PFK-2 stability is greatly decreased in the absence of insulin signaling. While we agree the transcript levels of PKA substrates may be affected by insulin, such an effect may be very selective. It is also worth noting that the phosphorylation of other substrates, such as phospholamban and troponin, were unaffected by insulin. 

4. In many cases, signaling intensity by Western blot is shown as a ratio of phosphorylated (active) form to total protein content. When the phosphorylated form decreases in diseased state and the total protein content also decreases, the ratio of phosphorylated form to total protein content is almost the same as that of control group. In this case, fold activation of diseased mice by β-adrenergic receptor stimulation will be about the same as control mice. When fold stimulation is same between control and diseased states, it cannot be concluded that decreased PKA signaling is due to the decreased expression of total protein content.

This is true and a limitation of the immunofluorescence technique. Nevertheless, we can tell that the response to a PKA agonist is different depending upon whether cardiomyocytes were in the presence or absence of insulin. I made it clear that we are referring to the phosphorylation of proteins and not total proteins in our immunofluorescence experiments. Furthermore, the limitations of the experiments are now more clearly stated. 

5. PKA signaling is mediated by signaling complex being comprised of PKA, A-kinase anchoring protein (AKAP), target protein, and other signaling proteins. Intensity, efficiency and specificity of PKA signaling depend on AKAPs in many cases. Therefore, the effects of insulin on decreased PKA signaling may be explained by the decreased expression of AKAPs. Western blot and immunofluorescence data of AKAPs expressing in the myocytes are necessary to substantiate the conclusion.

This is a good point. Changes in PKA may be due to alterations in its subcellular localization and mediated by AKAPs. Over 30 different AKAPs have been identified (https://doi.org/10.1016/j.cellsig.2017.05.012). It is unfortunately beyond the scope of this paper to determine if changes in AKAPs contribute to the insulin-dependent effects on PKA activity. We have previously reported though that there is no apparent change in AKAPs in the hearts of STZ model of type 1 diabetic mice using a binding assay (Bockus and Humphries, J. Biol. Chem., 2015, 290(49): 29250). In addition, we have now directly measured PKA activity (new Fig 1E). We show that the lack of insulin decreases basal PKA activity but not its activation by ISO. While we cannot rule out alterations in subcellular PKA, this supports that the primary defect is in discrete substrate phosphorylation.

6. When phosphorylation states is really wanted to be examined, phospho-proteomics analysis may be better than Western blot instead of using anti-PKA consensus sequence-antibody.

Phospho-proteomics is a powerful technique that will be used in future experiments, but it is beyond the scope of this work. 

7. Insulin has growth factor-like effects. Therefore, overnight culture without insulin may affect cell growth or maintenance. Thus, the effects of insulin on the isolated cells cultured overnight without insulin may be growth factor-like actions but not metabolic actions as suggested.

We agree that insulin has many effects on cellular function that extend beyond metabolism. We do not see any change in cell number or cell size with overnight +/- insulin treatment. However, we cannot rule out other effects and have added this caveat to the Discussion (lines 482-484). 

8. Insulin treatment may modify epigenetic modification during development of diabetes. It is interesting to examine insulin-induced epigenetic modification in myocytes of Akita diabetic model mice.

That is an interesting suggestion, but the epigenetic effects of insulin are beyond the scope of this work. 

Minor comments: 

1. At lines 456 to 458, authors described that ‘our results support that the deficiency of PKA signaling is not mediated by loss of β-adrenergic receptors, PKA protein, or cAMP production/degradation’. However, authors do not measure number of β-adrenergic receptors, and amounts of PKA protein and PDEs/adenylyl cyclases. It is too much to say that.

2. There are many proteins that their activities re regulated by insulin in myocytes. The decrease of PFK-2 level may not be enough to explain insulin-dependent metabolic changes in the heart.

1)The last paragraph of the Discussion was edited and this statement was removed. 2) We agree that insulin has many effects on cardiac metabolism such as increasing glucose uptake, activating glycolysis, and increasing glucose oxidation via the activation of pyruvate dehydrogenase. The novel aspect of this study is identifying that the lack of insulin affects PKA signaling. We have updated the Discussion to reflect other possibilities. 

Reviewer 2

1. There are several inconsistencies between the written text and what is shown in the figures. For example, line 191 onward describes WB data in Figure 1B, Figure 1B however, shows something different, not WB data. Continuing from that point, the authors state in line 199 "culturing myocytes in the presence of insulin for 18hrs had no enhancing effect on PKA substrate staining upon ISO stimulation". The authors need to be clear if they are comparing this to WT or baseline, as compared to baseline there is an effect with ISO stimulation in all groups. Far too much of my time was spent trying to interpret the data and I would urge the authors to proofread and ensure the manuscript is concise and to the point.

I apologize for the lack of clarity. The references to Fig 1 in the text now correspond to the figures accurately. In addition, we have changed the text extensively to improve clarity. Please note the highlighted regions throughout the Results section where it is made clear what groups are being compared.

2. No in vivo mouse data is provided in the manuscript. While the Akita model has previously been reported to be associated with diabetes, there is no data provided demonstrating that this model serves as a mechanism of low insulin as a result of diabetes in this study. It would be interesting to determine the overall insulin levels of Akita mice and how does this compare to WT without insulin.

The Akita mice are a well-established model of diabetes. The first study that characterized this model showed that mice are hypoinsulinemic and hyperglycemic. In vitro perfusion studies showed pancreatic beta cells do not release insulin in response to high glucose (Yoshioka et al. Diabetes, (1997) 46: 887).

I agree it would be interesting to see how well we are approximating insulin levels in Akita mice, in vivo, to WT ACMs without insulin. We unfortunately didn’t measure circulating insulin levels in these experiments. However, we did check blood glucose levels in the mice at the time of sacrifice. Blood glucose levels of all Akita mice were > 400 mg/dl. This information has been added to the Methods section. 

3. The use of immunofluorescence microscopy to monitor PKA signalling, as has been demonstrated in this study, has not been done before in adult myocytes. I am, therefore, surprised the authors have not validated this technique with other well-know measurements of PKA signaling such as a PKA activity kit to confirm the results. Immunofluorescence can often lead to false results; therefore the use of an antibody control is strongly advised. The manuscript would also benefit from clarification of how the mean fluorescence intensity images were normalized and the control measures taken to ensure there are no false positive signals.

Based on the reviewer’s suggestion, we have performed additional experiments to measure PKA activity in ACMs cultured in the presence or absence of insulin. This is now shown in Fig. 1E. Graduate student Maria Newhardt completed these experiments and has been added as an author. We report that basal PKA activity is decreased in wildtype ACMs cultured in the absence of insulin. Nevertheless, PKA was activated to a similar extent in the -/+ insulin conditions upon the addition of ISO. This supports that the robust insulin-dependent change in PKA substrate phosphorylation observed by immunofluorescence is likely due to changes in specific substrates and not a global defect in PKA activity. 

We agree that immunofluorescence data can be problematic. This is alleviated, in part, by using commercially sourced and validated antibodies. For the PKA substrate antibody, we have confidence in the results and specificity because of the responsiveness of the fluorescence signal to 3 different PKA activators: ISO, 8Br-cAMP, and PDE inhibitors. We also note that our experiments use the same secondary antibodies throughout the study. Staining was essentially undetectable in secondary antibody alone control experiments. This gives us confidence that the signal we are seeing is from the specificity of the primary. 

The confocal microscopy images were collected using the same settings between all experiments. This ensured reproducibility and allowed for unbiased comparisons between experimental conditions. This information has been added in the Materials and Methods section. MFI was calculated on maximum intensity projection images using the Zeiss Zen Black software. 

4. Extended time in culture leads to great changes in myocyte morphology. For example, the structural integrity of the cells, such as t-tubule organization becomes disrupted after prolonged culture. Both insulin receptors and B-receptors are located on the t-tubule membrane, thus it cannot be ruled out that signaling alterations could be a consequence of culture. Moreover, just the presence of insulin in the culture media may have metabolic benefits that prove more advantageous for long culture conditions. To address this, a time-course study would be useful, starting with the effects of insulin on PKA signaling in freshly isolated cells.

It is a limitation of ACMs that cells begin to dedifferentiate in culture. Our experiments our completed within 18h of isolation and we see no overt changes in morphology or viability when insulin is excluded from the media. 

Regarding the receptors, our experiments in Fig 7 show that reintroduction of insulin after 18h of culture results in robust phosphorylation of Akt. This would argue that our immunofluorescence results are not from decreased insulin receptors or their accessibility. Likewise, we see PKA signaling dysfunction when ACMs are cultured in the absence of insulin and are then stimulated with 8Br-cAMP or PDE inhibitors. 8Br-cAMP activates PKA directly and argues against a B-receptor defect. Changes in receptors cannot be ruled out, though. 

Regarding the time course, we have previously attempted such an experiment but it was subject to high variability. The reason, we believe, is that the cardiomyocytes are required to be in plating media containing horse serum when they are first isolated. This is then changed to serum-free culture media. We surmised that the variability arose from the persisting effects of the sera. For this reason, we chose 18h because cells sustained viability (regardless of -/+ insulin), morphology was not overtly different, it approximates a physiological fasted state, and the results were very consistent. 

5. There are concerns regarding the authors’ interpretation of the pPFK2 data. Especially, with the conclusion that PFK2 phosphorylation was decreased in an insulin-dependent manner. Although blunted compared with WT insulin hearts, WT hearts in the absence of insulin showed increased pPFK2 in response to ISO and IBMX. In the Akita hearts, however, ISO and IBMX failed to have any effect. This suggests that it is not just the absence of insulin, but something specific to the Akita mice leading to decreased pPFK2 activity.

I agree with the reviewer’s concern and I have modified the text to reflect other potential defects affecting PFK2 phosphorylation. I have also removed the sentence: “Thus, the immunofluorescence staining of phospho-PFK-2 follows closely to that of PKA substrate antibody.”

Minor issues

1. Were the stats performed on cells or animals, have the authors considered performing nested statistics to account for cells and animals?

We used Akita and WT mice that were, whenever possible, littermates. In addition, we attempted to minimize variability with our cell preparations by starting the experiment at the same time of day, using aliquots of reagents, and having the same personnel perform the experiments. I have consulted with a statistics expert, Albert Batushansky (see Batushanksy, Humphries, et al. Metabolomics 2019), and he confirmed our statistics are appropriate for the experimental design. 

2. It would be useful if the graphs displayed individual data points.

We believed the whisker plots were appropriate because it makes it easier for the reader to visualize differences between experimental conditions. Here is a representative plot with all the data points presented for the reviewer’s consideration (please see the response to reviewers in the PDF). We are willing to modify the figures or include all the data points in supplemental figures. 

3. The authors state PKA staining is found on the z-bands, but there is no mention as to which software was used to determine the position on the PKA activity staining?

The staining was largely diffuse but with apparent increased intensity along the membranes. There was also staining of the filaments that was qualitatively consistent with Z-bands. We have modified this statement.

---

## [Decision Letter · Decision Letter 1]

15 Jul 2020

PONE-D-20-09116R1

Diabetes induced decreases in PKA signaling in cardiomyocytes: the role of insulin

PLOS ONE

Dear Dr. Humphries,

Thank you for submitting your manuscript to PLOS ONE. After careful consideration, we feel that it has merit but does not fully meet PLOS ONE’s publication criteria as it currently stands. Therefore, we invite you to submit a revised version of the manuscript that addresses the points raised during the review process.

Your revised manuscript was reviewed by the original referees, and their comments are appended. As you will see they both recognize that the revised manuscript has adequately improved, while reviewer #2 pointed out "n" issue. The authors need to properly address this issue.

We look forward to receiving your revised manuscript.

Kind regards,

Makoto Kanzaki, Ph.D.

Academic Editor

PLOS ONE

Reviewers' comments:

Reviewer's Responses to Questions

**Comments to the Author**

1. If the authors have adequately addressed your comments raised in a previous round of review and you feel that this manuscript is now acceptable for publication, you may indicate that here to bypass the “Comments to the Author” section, enter your conflict of interest statement in the “Confidential to Editor” section, and submit your "Accept" recommendation.

Reviewer #1: All comments have been addressed

Reviewer #2: (No Response)

2. Is the manuscript technically sound, and do the data support the conclusions?

Reviewer #1: Yes

Reviewer #2: Partly

3. Has the statistical analysis been performed appropriately and rigorously? 

Reviewer #1: Yes

Reviewer #2: Yes

4. Have the authors made all data underlying the findings in their manuscript fully available?

Reviewer #1: Yes

Reviewer #2: Yes

5. Is the manuscript presented in an intelligible fashion and written in standard English?

Reviewer #1: Yes

Reviewer #2: Yes

6. Review Comments to the Author

Reviewer #1: (No Response)

Reviewer #2: Thank you for the opportunity to review this revised manuscript. The authors have thoroughly revised the paper to address my primary concerns. The authors have modified the text to provide clarity and clearly state the limitations of the study; making the study more accessible. Furthermore, adding additional experiments to measure PKA activity in ACMs make the results more robust and support the immunofluorescence data on insulin dependent change in PKA substrate phosphorylation.

At this stage, I have one minor concern, it is still not clear what exactly is N in this study. Are the statistics are performed on biological repeats (n=3) or cells (n= approx. 30) as is shown on the example data point plot provided? If it is biological repeats were the cell data averaged for each animal? On the contrary, if the stats were performed on cells, the authors need to be wary of pseudoreplication. It is recommended that a line is added to the methods section to clarify this.

7. PLOS authors have the option to publish the peer review history of their article (what does this mean?). If published, this will include your full peer review and any attached files.

Reviewer #1: No

Reviewer #2: No

---

## [Author Response · Author response to Decision Letter 1]

28 Jul 2020

We’d like to thank the reviewers for their evaluation and constructive critiques of our original and revised manuscripts. 

Reviewer 2: At this stage, I have one minor concern, it is still not clear what exactly is N in this study. Are the statistics are performed on biological repeats (n=3) or cells (n= approx. 30) as is shown on the example data point plot provided? If it is biological repeats were the cell data averaged for each animal? On the contrary, if the stats were performed on cells, the authors need to be wary of pseudoreplication. It is recommended that a line is added to the methods section to clarify this.

I apologize that this wasn’t clear. The experiments under a given condition were done on 3 cell preparations from 3 hearts. The statistics were performed on the sum of all the cells that were imaged. I have added additional information in the Methods section and in each of the Figure Legends to make this clear. I agree that n= appox. 30 can be considered as pseudoreplication, or as we considered them repeated measurements, and this could be a problem. However, when the cells from a given preparation are averaged and analyzed as n=3 similar conclusions are still reached. This information has been added in the “Statistical analysis” section in the Methods. For example, here is Figure 1 plotted with all the cell data shown or with the data plotted as averages of the 3 cell preparations. (These example Figures can be seen in our "Response to reviewers" in the PDF). We believe that showing the whisker plots encompassing all the cells more clearly demonstrates the data distribution and variability intrinsic to the method.

---

## [Decision Letter · Decision Letter 2]

7 Aug 2020

Diabetes induced decreases in PKA signaling in cardiomyocytes: the role of insulin

PONE-D-20-09116R2

Dear Dr. Humphries,

We’re pleased to inform you that your manuscript has been judged scientifically suitable for publication and will be formally accepted for publication once it meets all outstanding technical requirements.

Kind regards,

Makoto Kanzaki, Ph.D.

Academic Editor

PLOS ONE

Additional Editor Comments (optional):

Reviewers' comments:

Reviewer's Responses to Questions

**Comments to the Author**

1. If the authors have adequately addressed your comments raised in a previous round of review and you feel that this manuscript is now acceptable for publication, you may indicate that here to bypass the “Comments to the Author” section, enter your conflict of interest statement in the “Confidential to Editor” section, and submit your "Accept" recommendation.

Reviewer #1: All comments have been addressed

Reviewer #2: All comments have been addressed

2. Is the manuscript technically sound, and do the data support the conclusions?

Reviewer #1: Yes

Reviewer #2: Yes

3. Has the statistical analysis been performed appropriately and rigorously? 

Reviewer #1: Yes

Reviewer #2: Yes

4. Have the authors made all data underlying the findings in their manuscript fully available?

Reviewer #1: Yes

Reviewer #2: Yes

5. Is the manuscript presented in an intelligible fashion and written in standard English?

Reviewer #1: Yes

Reviewer #2: Yes

6. Review Comments to the Author

Reviewer #1: (No Response)

Reviewer #2: Thank you for the clarity and the additional information to address my concern regarding the statistics. All my comments have now been fully addressed.

7. PLOS authors have the option to publish the peer review history of their article (what does this mean?). If published, this will include your full peer review and any attached files.

Reviewer #1: No

Reviewer #2: No

---

## [Editor Report · Acceptance letter]

11 Aug 2020

PONE-D-20-09116R2 

Diabetes induced decreases in PKA signaling in cardiomyocytes: the role of insulin 

Dear Dr. Humphries:

I'm pleased to inform you that your manuscript has been deemed suitable for publication in PLOS ONE. Congratulations! Your manuscript is now with our production department. 

Kind regards, 

on behalf of

Dr. Makoto Kanzaki 

Academic Editor

PLOS ONE